

# Effects of shading on the growth and leaf photosynthetic characteristics of three forages in an apple orchard on the Loess Plateau of eastern Gansu, China

Mei Yang, Minguo Liu, Jiaoyun Lu and Huimin Yang

State Key Laboratory of Grassland Agro-ecosystems, College of Pastoral Agriculture Science and Technology, Lanzhou University, Lanzhou, China

## ABSTRACT

**Background**. Inclusion of forage into the orchard is of great help in promoting the use efficiency of resources, while shading from trees restricts forage growth and production in the Loess Plateau of China. This study was aimed to investigate how tree shading affected leaf trait, photosynthetic gas exchange and chlorophyll feature of forages under the tree in the orchard-forage system.

**Methods**. The shading treatments were set as partially cutting branches (reduced shading), normal fruit tree shading (normal shading) and normal tree shading plus sun-shading net (enhanced shading) in an apple orchard. Leaf trait, photosynthesis, chlorophyll component and fluorescence related parameters were measured with lucerne (*Medicago sativa*), white clover (*Trifolium repens*) and cocksfoot (*Dactylis glomerata*) which were sown under apple trees.

**Results**. Shading imposed significant impacts on the growth and leaf photosynthetic characteristics, while there were differences among species. Enhanced shading decreased leaf thickness, leaf dry matter content (LDMC) and leaf mass per unit area (LMA). Biomass accumulation decreased with enhanced shading in cocksfoot, but did not change in white clover and lucerne which had much lower biomass accumulation than cocksfoot. Enhanced shading reduced net photosynthetic rate (Pn) of white clover and lucerne, but rarely affected cocksfoot, while it decreased instantaneous water use efficiency (WUEi) of cocksfoot but had few effects on the other forages. Enhanced shading reduced leaf dark respiration rate (Rd), light compensation point (LCP) and maximum assimilation rate. The Rd and LCP of cocksfoot were much lower than those of white clover and lucerne. Chlorophyll contents and chlorophyll a/b changed little with shading. Cocksfoot had the highest contents but lowest ratio. Maximum photochemical rate of photosystem II increased and non-photochemical quenching decreased with enhanced shading in cocksfoot, while did not change in the other forages.

**Discussion**. Leaf trait, photosynthetic gas exchange and chlorophyll feature were variously affected by species, shading and their interaction. Cocksfoot was more efficient than the other two forages in use of weakened light and more tolerant to tree shading. In the apple orchard, we recommend that reducing the density of apple tree or partially cutting branches together with selecting some shading-tolerant forages, i.e., cocksfoot, would be a practical option for the orchard-forage system in the Loess Plateau of China.

Corresponding author
Huimin Yang, huimyang@lzu.edu.cn

# INTRODUCTION

Traditional orchard performance with bare ground or simple tillage to the soil has led to serious soil erosion and low use efficiency of resources, such as light, soil water and nutrients (*Shui et al., 2008*). Inclusion of grass (and/or forage) into the orchard is an advanced management mode for orchard soil (*Skroch & Shribbs, 1986*), offering a solution to deal with such issues, and has been widely used as an efficient conservation tillage in the orchard (*Neves et al., 2010*). However, the possible competition on soil water and nutrient between trees and grass under the trees has caused some worries that sowing grass in the orchard might result in loss of fruit yield and quality, especially in some areas where water deficit and soil infertility happen frequently (*Monteiro & Lopes, 2007*; *Teravest et al., 2010*), an example being the Loess Plateau of China. Therefore, the roles of grass in such an integrative system have attracted wide attention.

The inclusion of grass into the orchard can modulate soil features like water and fertility. Sowing grass may potentially adjust the enrichment and paucity of soil water content to keep it relatively stable (*Liu et al., 2013*). It can reduce surface runoff and enhance infiltration, alleviating soil erosion (*Fourie, Louw & Agenbag, 2007*). Moreover, with the increase of grass age, soil infiltration and water holding capacity will be greatly improved (*Palese et al., 2014*). There is competition for water between grass and tree, which varies with plant species and the amount of rainfall. The competition can be weak under suitable species combination and system management. Inclusion of grass can also be beneficial to improve the contents of organic matter (*Sánchez et al., 2007*), nitrogen (N), phosphorus (P) and potassium in soils (*Shui et al., 2008*). For instance, some legume species may potentially improve soil N availability as they have strong capacity to biologically fix atmospheric N (*Yang et al., 2011*). Additionally, soil microbial diversity and activity also increase in the orchard after sowing grass (*Whitelaw-Weckert et al., 2007*), which may be helpful for the decomposition of soil organism humus (*Wardle et al., 2001*). Therefore, under this system the competition for soil nutrient is relatively subtle due to the improvement of soil fertility by grasses and artificial fertilization. In addition, as the pattern of orchard-grass performance is continuously improved, transforming from firstly a single mode (ground cover) to the complex three–dimensional mode (combination of cover, farming and animal husbandry), the role of grass in the system is diversified. The grasses sown in the orchard may also be used as forages for feeding animals with countable amount of biomass accumulation in certain areas. Therefore, inclusion of grass into the orchard shows advantages both at ecological and economic scales.

The inclusion of grass has broken the water and heat exchange in soil—fruit tree—air continuum and has transformed into soil—fruit tree + grass—air continuum (*Bing et al., 2002*). In this way, water and heat can be more fully utilized in the system, which requires a balance between the growths of tree and grass to maximum their functions in

the system. However, tree shading may be a problem for the growth of grass under the tree as insufficient light causes adverse effects on grass growth and production. Generally, leaf net photosynthetic rate (Pn) may fall under shading and rapid stomatal closure occurs (*Kim, Oren & Qian, 2016*), while appropriate shading can improve water use efficiency (WUE) of plants, which varies with plant species. *Delucia et al. (1998)* found that the plants usually increased photosynthetic efficiency to improve light utilization by increasing leaf area under shading. With shading, leaf chlorophyll content increases and chlorophyll a/b value decreases to improve plant photosynthetic activity (*Abrams, 1987*; *Lambers & Poorter, 1992*). *Singhakumara, Gamage & Ashton (2003)* found that the shade-tolerant plants generally had larger leaf area, higher chlorophyll content and lower leaf mass per unit area (LMA) than the shade-sensitive ones. These aforementioned traits are important measures in plant adaptation to adverse light environments (*Grassi & Bagnaresi, 2001*) and thus may be helpful in selecting suitable grasses for the orchard. However, little knowledge has been achieved on how these traits of grass under the tree respond to shading in the orchard.

On the Loess Plateau of eastern Gausu, China, apple orchards are widely established as a profitable option in this arid and infertile area. There was approximately $1.02 \times 10^5$ ha apple orchards established in this area with apple yield of $6.7 \times 10^8$ kg per year. The existing orchards are mostly lightly tilled, which is unfavorable for controlling soil and water loss (*Wang et al., 2015*). Some traditional thoughts, i.e., grass and tree fight for water and nutrient in soils, and thus sowing grass may increase the costs of money and labour, have retarded the performance of grass sowing in the orchards of this area. Appropriate grass species are essential for the establishment of a sustainable orchard-grass system (*Wang et al., 2015*) but there was still rare species suitable for the system in this region. The lucerne (*Medicago sativa*), white clover (*Trifolium repens*) and cocksfoot (*Dactylis glomerata*) are common forage crops widely sown and used to feed domestic animals in this region. However, it was not clear how they can be better used in the orchard.

We proposed a hypothesis that tree shading would impose heavy impacts on grass species included in the orchard in some species-specific way. In this study, biomass, leaf trait, photosynthetic gas exchange and chlorophyll feature of three forages (grasses) were measured under three shading treatments in an apple orchard. The objectives were to find out: (1) how tree shading affects biomass accumulation, leaf trait, photosynthetic gas exchange and chlorophyll feature of the forages? (2) Which of the three species is more tolerant to shading in the orchard?

## MATERIALS & METHODS

### Plant material and experimental design

The experiment was conducted in a 7 year–old apple orchard at Qingyang Loess Plateau Pastoral Agriculture Station of Lanzhou University (35°40′N, 107°51′E), which locates in Qingyang, eastern Gansu of China with a typical continental climate. The mean annual precipitation is 543 mm and 70% of this total usually falls in July to September. The mean annual temperature is 9.3 °C with the lowest in January (−21.3 °C) and the highest in July

(40 °C). The annual frost-free duration is 255 d in average. The soil is Heilu soil with 70% silt and 23% clay, representing the main cropping soil in this area.

In the intervals between tree lines (4 m wide), three forages were broadcast sown, which are lucerne (*Medicago sativa*), white clover (*Trifolium repens*) and cocksfoot (*Dactylis glomerata*) in July 4, 2014. The sowing rates were 22.5, 15.0 and 15.0 kg ha$^{-1}$ for lucerne, white clover and cocksfoot respectively. For this test, 6 m long (×4 m wide) plots were chosen and for each treatment, four replicates were set. All plots were broadcast applied with 300 kg ha$^{-1}$N fertilizer in the form of urea before sowing. The forage was supposed to be cut and the shoot was removed out so that great amount of nutrient (especially N) would be lost from the system, so N fertilizer was applied, even to legume species. Soil P in the orchard was excessive due to long term P fertilization and slow release of soil P source and the inclusion of forages would benefit the release of soil residual P in the orchard, so P fertilizer was not applied. No irrigation and pesticide spraying were performed. All the forages were cut to feed domestic animals after plant samples taken. Notably, no treatment and measurement were conducted in the first year in order to favour the establishment of grasslands under the trees.

The shading treatments were started in April 12 before the forages were reviving in the second year. Three shading treatments were set as reduced shading (partially cutting branches), normal shading (normal tree shading) and enhanced shading (normal tree shading plus sun-shading net), and these treatments made the light intensity equal to about 70%–80%, 40%–50% and 10%–20% photosynthetically active radiation (PAR) above the canopy, which we measured every 2 weeks on sunny days using a portable photosynthesis system (LI–6400, Li–Cor, USA). All measurements were conducted at about two months later after shading treatment when lucerne and white clover were at early flowering stage and cocksfoot was at late heading stage. Due to budgetary limit, all measurement was only conducted in this year. Considering all three species are perennials and the second year is very close to the stabilized ages of artificial grassland in this area, the data we obtained should show the characteristics of the second year's forages.

## Measurements and calculations

At least 20 youngest fully expanded leaves were sampled for each treatment. The sampled leaves were then brought back to laboratory as soon as possible for further measurements. Leaf biomass at saturated moisture content and dry weight were measured to determine leaf dry matter content (LDMC) (*Garnier et al., 2001*) using the equation: LDMC (mg g$^{-1}$) = leaf dry weight/leaf saturated fresh biomass. Fresh leaf area (cm$^2$) was scanned with Win FOLIA (LA2400, Canada) and the LMA was determined using the equation: LMA (g m$^{-2}$) = leaf dry weight/leaf area. In addition, leaf thickness (LT) was measured with a vernier caliper. Biomass was measured with drying method. After sampling with quadrat frame of 1 m × 1 m, the samples were dried at 80 °C until constant weight and measured the biomass on the ground.

Constant photosynthetic gas exchange was measured with a portable photosynthesis system (LI–6400, Li–Cor, USA) at 9:30–11:30 am on a clear sunny day during leaf sampling. The $CO_2$ concentration was maintained at 400 µL L$^{-1}$ using $CO_2$ supplying cartridge.

The Pn ($\mu$mol m$^{-2}$ s$^{-1}$) and transpiration rate (E, mmol m$^{-2}$ s$^{-1}$) were recorded and instantaneous WUE (WUEi, $\mu$mol mmol$^{-1}$) was calculated as Pn/E. In each replicate, three plants were selected randomly and at least three healthy and fully expanded leaves were measured. The Pn response to light gradient was measured at 09:00–11:00 on a clear sunny day using the red and blue light source equipped with LI–6400. During the measurements, $CO_2$ concentration was maintained at 400 $\mu$L L$^{-1}$ using $CO_2$ supplying cartridge and light intensity was set according to the Equipment Instruction. The curve was then fit with the classic Farquhar model (*Farquhar, Caemmerer & Berry, 2001*) to obtain light compensation point (LCP), light saturation point (LSP), dark respiration rate (Rd), maximum assimilation rate (Amax) and apparent quantum efficiency (Qapp).

Chlorophyll a and b was extracted by mixture of propanone and anhydrous ethyl alcohol, and then the contents were determined by spectrophotometer method of *Arnon (1949)*. The contents of chlorophyll a, b and a+b were calculated using the following equations:

Chlorophyll a (mg g$^{-1}$) = [(12.7 × A663 − 2.59 × A645)V/W],

Chlorophyll b (mg g$^{-1}$) = [(22.9 × A645 − 4.67 × A663)V/W],

Chlorophyll a+b (mg g$^{-1}$) = [(20.3 × A645 + 8.04 × A663)V/W],

Where A is absorbance at specific wavelengths; V is final volume of chlorophyll extract; W is fresh weight of leaf extracted. In the present experiment, the volume (V) and weight (W) were 100 ml and 0.1 g, respectively.

Chlorophyll fluorescence was measured at 09:00–11:00 on a clear sunny day to obtain actual photochemical efficiency of photosystem II ($\phi$PS II), photochemical quenching coefficient (qP) and non-photochemical quenching (NPQ) using fluorescent leaf chamber of LI–6400 with controlled light intensity of 1,500 $\mu$mol m$^{-2}$ s$^{-1}$. Prior to these measurements, marked leaves were measured in dark to determine maximum photochemical rate (Fv/Fm) at 01:00 deep night. In each replicate, three plants were selected randomly and at least three healthy and fully expanded leaves were measured.

## Data analysis

The effects of shading treatment, forage species and their interaction on leaf trait, gas exchange, chlorophyll component and fluorescence were analyzed using factor analysis. The differences in leaf traits, gas exchange parameters and chlorophyll features among forages or shading treatments were analyzed using Two–Way ANOVA. The Pn–PAR curve was fit with the classic Farquhar model to gain related parameters. Correlations among the growth and leaf photosynthetic characteristics of three forages under shading were analyzed using Spearman's rank correlation analysis. The SPSS 17.0 was used for statistical analysis.

## RESULTS

### Leaf traits and biomass growth under shading

The LDMC and LMA of three forages were affected by shading, species and their interaction, while the LT was only affected by shading and the biomass was only affected by species (Table 1). The LT of cocksfoot did not change under all treatments but it tended to decrease with the enhancement of shading. The LT of white clover decreased with the

**Table 1  Effects of species, shading and their interaction on leaf trait, photosynthetic gas exchange and chlorophyll feature in the orchard-forage system.**

| Factor | Biomass | LT | LDMC | LMA | Pn | WUEi | Chlorophyll a | Chlorophyll b | Chlorophyll a +b | Chlorophyll a/b | Fv/Fm | $\phi$PS II | qP | NPQ |
|---|---|---|---|---|---|---|---|---|---|---|---|---|---|---|
| Species | *** | NS | *** | ** | ** | *** | *** | *** | *** | *** | *** | *** | *** | *** |
| Shading | NS | * | *** | *** | *** | *** | NS | NS | NS | * | NS | NS | NS | NS |
| Species × Shading | NS | NS | * | *** | *** | *** | * | NS | * | NS | ** | ** | ** | ** |

Notes.

NS, non-signicant; *, signicance at $P \leq 0.05$; **, signicance at $P \leq 0.01$; ***, signicance at $P \leq 0.001$; LT, leaf thickness; LDMC, leaf dry matter content; LMA, leaf mass per unit area; Pn, net photosynthetic rate; WUEi, instantaneous water use efficiency; Fv/Fm, maximum photochemical rate; $\phi$PS II, actual photochemical efficiency of II; qP, photochemical quenching coefficient; NPQ, non-photochemical quenching.

enhancement of shading. For lucerne, LT was reduced by enhanced shading and did not change under reduced shading (Fig. 1A). The LDMC of cocksfoot and lucerne decreased with the enhancement of shading. For white clover, LDMC was reduced by enhanced shading and was not changed by reduced shading (Fig. 1B). The LMA of all three forages decreased with the enhancement of shading (Fig. 1C). In response to the enhancement of shading, LT of cocksfoot did not change, but LDMC and LMA decreased, and its LMA was the lowest among all forages under enhanced shading (Fig. 1). The biomass of cocksfoot was much higher than the other forages, which tended to decrease with enhanced shading. The biomass of white clover and lucerne were not changed by shading (Fig. 1D).

## Leaf photosynthetic gas exchange under shading

The Pn and WUEi were affected by shading, species and their interaction (Table 1). The Pn of cocksfoot was reduced by enhanced shading and did not change under reduced shading. The Pn of white clover and lucerne decreased with the enhancement of shading (Fig. 2A). The WUEi of cocksfoot decreased with the enhancement of shading. For whiter clover and lucerne, WUEi was elevated by enhanced and reduced shading (Fig. 2B). In response to the enhancement of shading, the Pn of cocksfoot changed little and both Pn and WUEi were not different from those of other two forages under enhanced shading (Fig. 2).

The shading imposed various impacts on Pn–PAR curves of different forages (Table 2). The Rd of cocksfoot was elevated by reduced shading but was not impacted by enhanced shading. The Amax and LCP of cocksfoot tended to decrease with the enhancement of shading, while LSP was reduced by enhanced and reduced shading but Qapp was elevated. The Rd of white clover was reduced by enhanced and reduced shading. The Amax, LCP and LSP of white clover tended to decrease with the enhancement of shading. Its Qapp tended to decrease under enhanced and reduced shading. For lucerne, the Rd, Amax and LCP tended to decrease with the enhancement of shading. The LSP tended to decrease under enhanced and reduced shading but the Qapp was elevated. The Rd and LCP of cocksfoot were lower than other forages but the Amax was not different and the Qapp was even higher under enhanced shading.

## Leaf chlorophyll component and fluorescence under shading

The content and proportion of chlorophyll component were affected by species, but seldom by shading and their interaction (Table 1). For cocksfoot, the contents of all chlorophyll

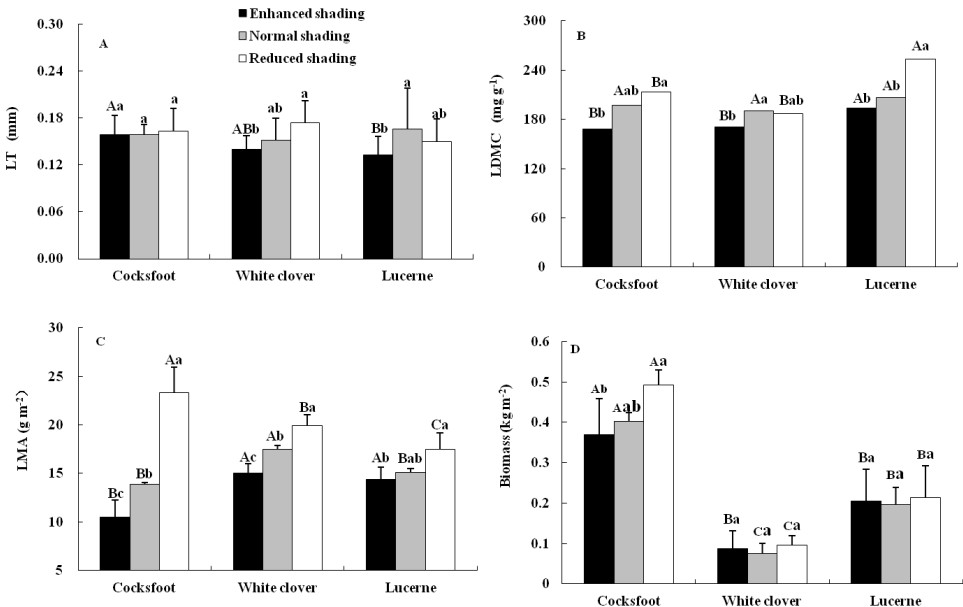

**Figure 1  Leaf thickness (LT) (A), leaf dry matter content (LDMC) (B), leaf mass per unit area (LMA) (C) and biomass (D) of three forages under shading.** Different capital letters denote significant differences among species under the same shading treatment ($P \leq 0.05$). Different lowercase letters denote significant differences among shading treatments for the same species ($P \leq 0.05$). Bars show standard deviation.

components and chlorophyll a/b were not impacted by enhanced and reduced shading (Fig. 3). For white clover, the contents of chlorophyll a, b and a+b increased with the enhancement of shading, while chlorophyll a/b decreased. For lucerne, the contents of all chlorophyll components and chlorophyll a/b were not changed by enhanced and reduced shading (Fig. 3). The contents of chlorophyll a, b and a+b were highest in cocksfoot, while for chlorophyll a/b, it appeared as white clover>lucerne>cocksfoot (Fig. 3).

Chlorophyll fluorescence was affected by species and their interaction (Table 1). The Fv/Fm increased with the enhancement of shading in cocksfoot and was not changed in white clover, while in lucerne, Fv/Fm was elevated by enhanced and reduced shading, and it was higher under reduced shading than enhanced shading (Fig. 4A). The φPS II and qP of cocksfoot were lowered by enhanced and reduced shading and in white clover they were not impacted, while in lucerne, the φPS II and qP decreased with the enhancement of shading (Figs. 4B and 4C). In cocksfoot, the φPS II and qP were higher under enhanced shading than reduced shading. The NPQ of cocksfoot decreased with the enhancement of shading. For white clover, NPQ was lowered by enhanced and reduced shading, while in lucerne, NPQ was elevated (Fig. 4D). In response to the enhancement of shading, the Fv/Fm increased in cocksfoot and its NPQ was far lower than those in other two forages (Figs. 4A and 4D).

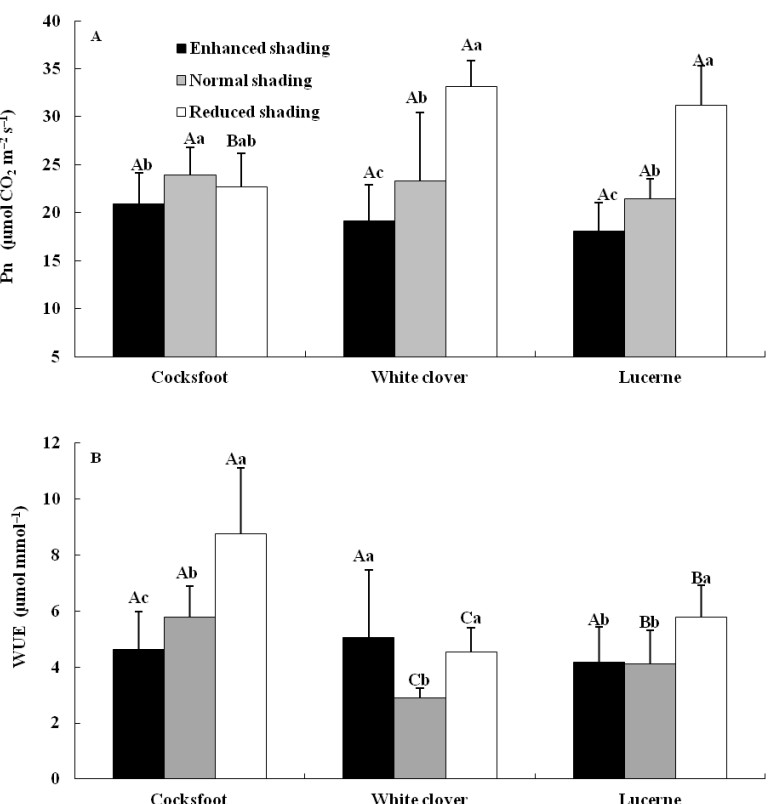

**Figure 2** **Net photosynthetic rate (Pn) (A) and instantaneous water use efficiency (WUEi) (B) of three forages under shading.** Different capital letters denote significant differences among species under the same shading treatment ($P \leq 0.05$). Different lowercase letters denote significant differences among shading treatments for the same species ($P \leq 0.05$). Bars show standard deviation.

**Table 2** Optimized parameters of the exponential rise to max function from Pn-PAR curves of three forages under shading.

| | Cocksfoot | | | White clover | | | Lucerne | | |
|---|---|---|---|---|---|---|---|---|---|
| | Enhanced shading | Normal shading | Reduced shading | Enhanced shading | Normal shading | Reduced shading | Enhanced shading | Normal shading | Reduced shading |
| Rd ($\mu$mol m$^{-2}$ s$^{-1}$) | 0.35 | 0.33 | 1.54 | 1.22 | 1.60 | 1.29 | 1.24 | 2.21 | 3.28 |
| Qapp ($\mu$mol mol$^{-1}$) | 0.07 | 0.04 | 0.10 | 0.06 | 0.07 | 0.05 | 0.06 | 0.05 | 0.06 |
| LCP ($\mu$mol m$^{-2}$ s$^{-1}$) | 5.1 | 8.3 | 15.9 | 21.3 | 24.1 | 26.7 | 21.4 | 46.8 | 58.7 |
| LSP ($\mu$mol m$^{-2}$ s$^{-1}$) | 722 | 1757 | 707 | 524 | 544 | 1079 | 814 | 880 | 819 |
| Amax ($\mu$mol m$^{-2}$ s$^{-1}$) | 13.8 | 17.7 | 25.3 | 13.2 | 16.4 | 19.8 | 18.2 | 20.0 | 24.5 |

**Notes.**
Rd, dark respiration rate; Qapp, apparent quantum efficiency; LCP, light compensation point; LSP, light saturation point; Amax, maximum assimilation rate.

## Correlations among the growth and leaf photosynthetic characteristics of three forages under shading

The biomass was positively correlated with WUEi, chlorophyll a, b and a+b contents, but negatively correlated with chlorophyll a/b, Fv/Fm, φPS II, qP and NPQ (Table 3). The Pn was positively correlated with LDMC, LMA, chlorophyll a/b, Fv/Fm, φPS II and qP, but

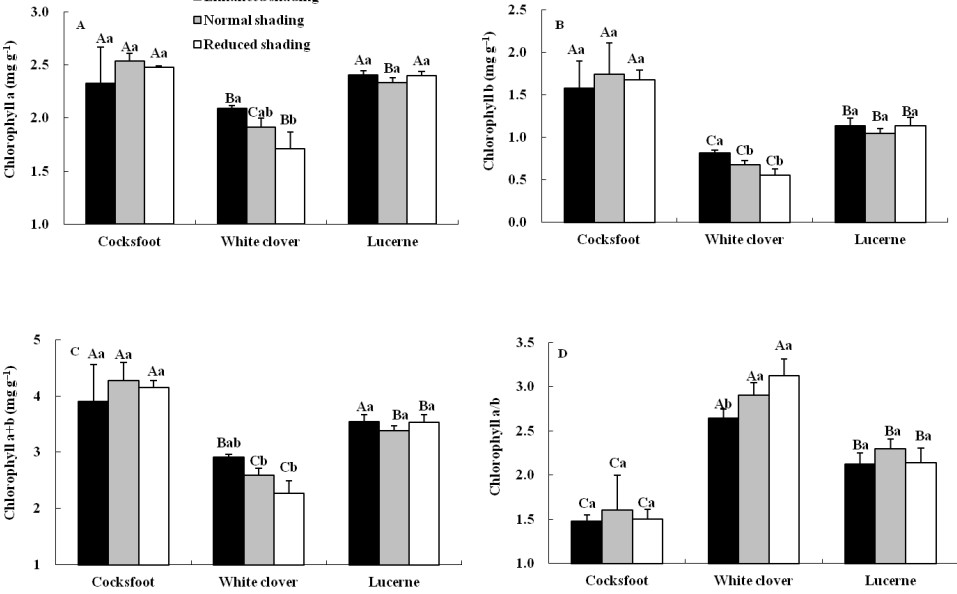

**Figure 3** **Chlorophyll a (A), b (B), a+b (C) and a/b (D) in leaves of three forages under shading.** Different capital letters denote significant differences among species under the same shading treatment ($P \leq 0.05$). Different lowercase letters denote significant differences among shading treatments for the same species ($P \leq 0.05$). Bars show standard deviation.

negatively correlated with chlorophyll a+b content. The WUEi was positively correlated with LDMC, LMA, chlorophyll a, b and a+b contents, but negatively correlated with chlorophyll a/b, Fv/Fm, φPS II, qP and NPQ.

## DISCUSSION

### Effects of shading on leaf traits of three forages

Leaf trait is partly the consequence that a plant responds to the external environments at leaf scale (*Vendramini et al., 2002*) and its change is one of the most important strategies that the plant has developed to cope with adverse environments. This study showed that species, shading and their interaction imposed significant impacts on leaf traits. Shading reduced LT, LDMC and LMA. Thus, shading may reduce assimilates accumulation but enhance the allocation for potential enhancement of photosynthetic photon capture because lower LT, LDMC and LMA generally indicate more input into photosynthetic area (*Modrzy et al., 2015*). These changes would result in enhanced photosynthesis. In addition, shading may help to maintain soil water status and improve air humidity under the tree, which potentially ameliorate the possible water stress that the forages are encountering, especially in this semi-arid and rainfed region. The LT was not affected by species, but it tended to be higher in cocksfoot than other species and did not change with shading, suggesting that cocksfoot may be more tolerant to shading. Much quicker decline in LMA of cocksfoot also proved that this species is more adaptive to shading, as lower LMA shows stronger potential to use weak light under the tree.

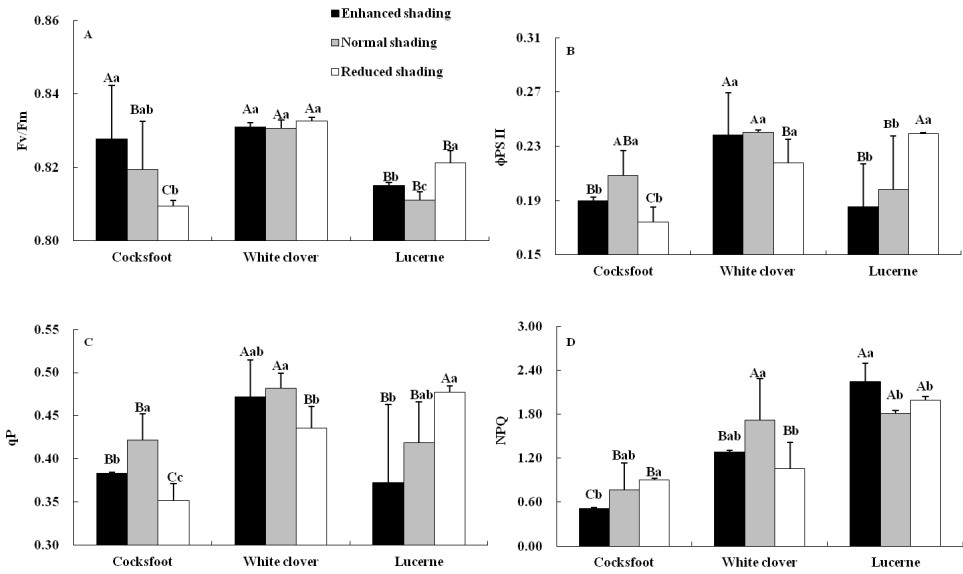

**Figure 4  Maximum photochemical rate (Fv/Fm) (A), actual photochemical efficiency of photosystem II (φPS II) (B), photochemical quenching coefficient (qP) (C) and non-photochemical quenching (NPQ) (D) in leaves of three forages under shading.** Different capital letters denote significant differences among species under the same shading treatment ($P \leq 0.05$). Different lowercase letters denote significant differences among shading treatments for the same species ($P \leq 0.05$). Bars show standard deviation.

**Table 3  Correlations among the biomass, leaf thickness (LT), leaf dry matter content (LDMC), leaf mass per unit area (LMA), net photosynthetic rate (Pn), instantaneous water use eciency (WUEi), chlorophyll content and fluorescence of three forages under shading.**

| | LT | LDMC | LMA | Pn | WUEi | Chlorophyll a | Chlorophyll b | Chlorophyll a+b | Chlorophyll a/b | Fv/Fm | φPS II | qP | NPQ |
|---|---|---|---|---|---|---|---|---|---|---|---|---|---|
| Biomass | 0.06 | 0.14 | 0.04 | −0.15 | 0.49*** | 0.54*** | 0.79*** | 0.73*** | −0.79*** | −0.52*** | −0.60*** | −0.63*** | −0.49*** |
| Pn | 0.11 | 0.30** | 0.31** | 1 | 0.30** | −0.17 | −0.19 | −0.19* | 0.24* | 0.25** | 0.29** | 0.32*** | −0.05 |
| WUEi | 0.04 | 0.22* | 0.38*** | 0.30** | 1 | 0.23* | 0.39*** | 0.34*** | −0.39*** | −0.29** | −0.32*** | −0.33*** | −0.31** |

**Notes.**

Spearman's correlation coefficients ($n = 108$) are shown.

***, ($P \leq 0.001$); **, ($P \leq 0.01$); *, ($P \leq 0.05$); Fv/Fm, maximum photochemical rate; φPS II, actual photochemical efficiency of PS II; qP, photochemical quenching coefficient; NPQ, non-photochemical quenching.

## Effects of shading on chlorophyll contents and fluorescence of three forages

Generally, a plant with high chlorophyll content and low chlorophyll a/b has stronger resistance to shading (*Boardman, 1977*). In this study, species and its interaction with shading significantly affected chlorophyll a, b, a+b contents and a/b, while shading showed rare effect. Only in white clover, the contents reduced and chlorophyll a/b increased along with the reduction in shading, suggesting that chlorophyll content and ratio weren't influenced by shading in cocksfoot and lucerne. Intriguingly, the contents of chlorophyll component in cocksfoot were highest, but chlorophyll a/b was lowest, indicating that

cocksfoot is more efficient in use of weak light because high chlorophyll b content and proportion promise a plant to do so (*Abrams, 1987*; *Threlfall, 1981*).

Chlorophyll fluorescence reflects the actual and maximum photosynthesis, the function of reaction center and the heat dissipation of a plant (*Govindjee, 2002*). In this study, all fluorescence was significantly affected only by species. The NPQ tended to be lowest in cocksfoot, and the Fv/Fm increased with the enhancement of shading, suggesting that this species is more tolerant to shading. The increased Fv/Fm reflects the enhancement of potential PSII photochemical efficiency of leaves after a fully dark adaptation (*Demmig & Björkman, 1987*). The lower NPQ shows less light energy consumption as heat dissipation (*Genty, Briantais & Baker, 1989*). Compared to lucerne and white clover, cocksfoot showed normal light conversing efficiency and light trapping efficiency under shading, but much lower light energy loss, helping to adapt to weakened light environments.

## Effects of shading on Pn, WUEi and biomass of three forages

In this study, constant Pn and WUEi were significantly affected by species, shading and their interaction. The Pn and WUEi tended to decrease with the enhancement of shading. These may suggest that under tree shading, weakened light led to Pn decrease because generally, Pn and light intensity are positively correlated with suitable water supply under natural light. However, improved water status in soils and relative air humidity under the trees would have kept stomata open, consequently leading to great transpiration (rate). Thus, the WUEi would decrease with shading as it was calculated with Pn/E. From another viewpoint, it also proved that shading may improve water supply around the forage and tree. Changes in Pn with shading among species may be due to variations in leaf traits and chlorophyll features as there were positive correlations of Pn with LDMC, LMA and chlorophyll a/b, $\phi$PS II and qP, and negative correlation with chlorophyll a+b content. As for WUEi, there were contrasting roles played by chlorophyll features as the WUEi was positively correlated with chlorophyll a, b and a+b contents, but negatively correlated with chlorophyll a/b, Fv/Fm, $\phi$PS II, qP and NPQ. Compared with other forages, the Pn of cocksfoot changed little with shading, and both Pn and WUEi were not different from other forages, suggesting that this species was more tolerant to shading.

The Pn–PAR curve provides very useful parameters to address photosynthetic responses of a plant to adverse environment, while eliminating much interference, i.e., insufficient light radiation (*Lewis et al., 2000*). In this study, the LCP and Amax decreased with the enhancement of shading, suggesting that all species are acclimating to shading, while the Rd, Qapp and LSP changed in a species-specific way. Compared to white clover and lucerne, the Rd and LCP of cocksfoot were lower, but the Amax was similar, and the Qapp was even higher under shading, indicating that cocksfoot may be more tolerant. The lower LCP indicates that the plant can survive in weakened light environments (*Taiz & Zeiger, 2010*), i.e., tree shading, and generally, shading-tolerant plants have lower Rd (*Lewis et al., 2000*). The greater Qapp reflects stronger photosynthesis to use weak light. Thus, it suggested that cocksfoot could make better use of weak light and adapt to shading, compared to other species.

The biomass of cocksfoot was much higher than the other forages, which decreased little under enhanced shading compared to normal shading. Cocksfoot has good adaptability to various environmental conditions, such as drought and restricted light conditions, with good regrowth characteristics (*Sanada, Gras & Santen, 2010*). Change in biomass accumulation with shading among species was more correlated with WUEi, but not Pn in the orchard environment. Therefore, cocksfoot might be more beneficial to provide biomass under tree shading.

It is known that plant biomass accumulation was not only impacted by light, but also by soil carbon and nutrients. Soil nutrients (such as N and P) can indirectly affect the utilization of light radiation by regulating photosynthesis apparatus (*Arain et al., 2002*; *Palmroth et al., 2014*). In this study, there were similar basic soil feature and relative enough nutrient supply to soils. Therefore, the difference in effects of soil nutrients on the plant might be negligible. However, it is obliged to admit the fact soil nutrient availability would change after longer time forage growth and this would affect the response of forage to light radiation, so further studies would be required in the future.

## CONCLUSIONS

Shading imposed significant impacts on the growth and leaf photosynthetic characteristics, while there were differences among species. Shading affected chlorophyll content and fluorescence, LDMC and LMA, which finally changed biomass accumulation. Cocksfoot was more efficient than the other two forages in use of weak light and more tolerant to tree shading. In the apple orchard, we recommend that selecting some shading-tolerant grasses, i.e., cocksfoot, or widening the distance between individual apple trees, would be practical options for the orchard-forage system in the Loess Plateau of China.

## ACKNOWLEDGEMENTS

We appreciate very much the assistance from Mr. Binghong Duan, Ms. Yaya Wang, Dr. Qian Yang and Mr. Juncheng Li in the field work and in the lab measurement and the comments on revising this draft from Prof. Yuying Shen and Dr. Ran Xue.

### Funding

This work was jointly supported by the grants from the National Science and Technology Support Program (2014BAD14B006) and the National Natural Science Foundation of China (31572460). The funders had no role in study design, data collection and analysis, decision to publish, or preparation of the manuscript.

### Grant Disclosures

The following grant information was disclosed by the authors:
National Science and Technology Support Program: 2014BAD14B006.
National Natural Science Foundation of China: 31572460.

## Competing Interests

The authors declare there are no competing interests.

## Author Contributions

- Mei Yang conceived and designed the experiments, performed the experiments, analyzed the data, prepared figures and/or tables, authored or reviewed drafts of the paper, approved the final draft.
- Minguo Liu and Jiaoyun Lu performed the experiments.
- Huimin Yang conceived and designed the experiments, authored or reviewed drafts of the paper, approved the final draft.

## Data Availability

The raw data are available as a Supplemental File. The raw data shows leaf trait, photosynthesis, chlorophyll component and fluorescence related parameters of lucerne (Medicago sativa), white clover (Trifolium repens) and cocksfoot (Dactylis glomerata) which were sown under apple trees.

## Supplemental Information

Supplemental information for this article can be found online at http://dx.doi.org/10.7717/peerj.7594#supplemental-information.

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
