# Peer review of "Effects of shading on the growth and leaf photosynthetic characteristics of three forages in an apple orchard on the Loess Plateau of eastern Gansu, China"

_PeerJ, doi:10.7717/peerj.7594_

## Round 0.1 · original submission · Major Revisions

The reviewers gave critical comments demanding manuscript revisions. It is rather small revision, but I recommend major revisions to give you an opportunity to fix all the remarks. The topic is of interest. I encourage you to re-submit the manuscript to PeerJ after revision.

Reviewer 1 ·

Basic reporting

no comment

Experimental design

no comment

Validity of the findings

no comment

Additional comments

Inclusion of forage crops into the orchard is advantageous to gain both ecological and economic goals. This work focused on the photosynthetic and morphological properties of three forages under apple tree. There were interesting results found. Each experiment was well designed. However, for more benefits of readers, a number of points need clarifying, and certain statements require further explanation.
How widely is the apple orchard established in the testing area and how about orchard-forage pattern? Do pay attention to the terms “forage”, “grass” in this part.
In the introduction, you mentioned the modulation of water and fertility of soils with forage sowing. How forage inclusion affected soil water? Under what condition the possible competition for water would no longer be a problem for both trees and forages?
In the materials and methods, the author should tell how the forage species were managed, by cutting? Grazing? Or they were commonly used in this region only as soil cover? Try explaining why there wasn’t P fertilization?
The results were mostly assigned to the different shading and potential effects with soil water and nutrient availability were not well considered. More information should be given on the soil water and nutrients which might have contributed to the findings. The information can be addressed in the discussion or introduction.
Why you stated to widen the distance between individual trees? Please clarify the reason to reduce the density of apple tree. If these forage species will be used just as soil cover, this density of trees may be good enough.
Line 41: Change “In the apple yard” into “In the apple orchard”.
Line 53: Delete “beneath”.
Line 148: “degrees” be replaced by ℃.
And other minor English issues...

Reviewer 2 ·

Basic reporting

Line 42, why reducing density of apple trees? In the apple yard, the goal is growing apple trees not grass, right? Planting grass is to help reduce soil erosion and enrich soil fertility for trees to grow.

Line 56, I suggested authors not introduce this issue/question, as this paper is not studying interactions between grass and trees or how grass impacts tree growth (competition).

Line 100, change this sentence "..they can be used.." to "..better being used?". I am sure all those species can be used.

Line 104, "under three shading treatments"

Line 106, change the second question to be more conservative, "which of the three species is more tolerant to shading in the orchard". The decision of which is more suitable involves more considerations (e.g. economy, diversity, etc). And usually we spell it out if count is less than 10, so use "three" instead of "3".

Table 2, it is better to add SE/SD to those values since authors had three replicates.

For Figure captions, use "Normal shading" would be better for "shading"

Experimental design

Line 112, "The mean annual precipitation is 543 mm with... The mean annual temperature.."

Line 147, would not 80 degrees too high for drying? I usually see people dry plant tissues around 55-60. Possible high drying temperature would cause loss of nitrogen

Line 170-176, put one sentence in the end "SPSS 17.0 was used for statistical analysis", so authors can delete the 3 extras.

Validity of the findings

Overall for Result section, 1. I suggested authors to put p values next to the sentences that describing the statistical results including differences between treatments and correlation test, so it would help readers to understand when authors said "hardly" or "barely". 2. Within each paragraph, please keep the subject consistent. Authors started with measured parameters such as "LDMC, LMA" as a subject in line 180, but used treatments such as "enhanced shading.." as a subject in line 183. 3. And for sentences describing insignificant results, use "did not influence.." or "was not changed/impacted" or "was similar" instead of "hardly affected it"

Line 215, the measurement of chlorophyll a and chlorophyll b was not mentioned in the Method section. And authors should add more details/explanations in the Data analysis part about why using chlorophyll a+b and a/b but not chlorophyll a or b alone?

I think in the Discussion section, authors should mention the soil nutrient status for the three forages if they had those data. Because some measurements here (e.g. biomass) is not only impacted by light but also soil carbon and nutrients. How the differences of soil nutrients between those plots bias the results of shading effects?

Additional comments

This paper studied the effects of three shading treatment on Leaf trait, photosynthesis, chlorophyll component and fluorescence for three grass species including lucerne (Medicago sativa), white clover (Trifolium repens) and cocksfoot (Dactylis glomerata). It reported that shading usually decreases leaf thickness, leaf dry matter content, leaf mass per unit area and net photosynthetic rate. Shading changed little to Chlorophyll contents. Among three species, cocksfoot had the least impacts compared to white clover and lucerne. I found this study is valuable but should be improved based on the above comments before acceptance.

---

## Round 0.2 · Minor Revisions

This manuscript version got positive estimates. However there are some minor comments. Please fix it soon and resubmit the manuscript. I think the paper could be published without next review round after the update.

Reviewer 1 ·

Basic reporting

The revised paper is clear and readily understood. Sufficient references were cited. It is well structured and the figures and tables were of high quality. Raw data were shared. And the hypotheses were well tested.

Experimental design

The experiment is well designed and the methods are right.

Validity of the findings

The results were presented with good analyses. The conclusion is well stated.

Additional comments

The authors made a thorough revision and responded to the comments in a suitable way. In my opinion, it can be accepted at its present form.

Reviewer 2 ·

Basic reporting

None

Experimental design

None

Validity of the findings

None

Additional comments

I really appreciated authors' hardworking on revising this paper. I only have two minor question: 1."Why change all the subheading to Arial but not Times New Rome?" 2. Line 234, no need to mention those values are constant, omit "12.7, 2.59, 22.9, 4.67, 20.3 and 8.04 are the constants".

---

## Round 0.3 · accepted · Accept

The manuscript had only minor remarks from second reviewer at previous review round. Now he has no critical comments. I believe the paper is ready for publication.

Reviewer 2 ·

Basic reporting

None

Experimental design

None

Validity of the findings

None

Additional comments

I find this paper can be accepted as it is!